# A Novel Rodent Model of Hypertensive Cerebral Small Vessel Disease with White Matter Hyperintensities and Peripheral Oxidative Stress

**DOI:** 10.3390/ijms23115915

**Published:** 2022-05-25

**Authors:** Reut Guy, Rotem Volkman, Ella Wilczynski, Chana Yagil, Yoram Yagil, Michael Findler, Eitan Auriel, Uri Nevo, Daniel Offen

**Affiliations:** 1Felsenstein Medical Research Center, Department of Human Genetics and Biochemistry, Sackler School of Medicine, Tel Aviv University, Tel Aviv 69978, Israel; v.rotem@gmail.com (R.V.); findlerm@gmail.com (M.F.); eitanman1@gmail.com (E.A.); danioffen@gmail.com (D.O.); 2Department of Biomedical Engineering, The Iby and Aladar Fleischman Faculty of Engineering, Tel Aviv University, Tel Aviv 69978, Israel; ellaalon@mail.tau.ac.il (E.W.); nevouri@tauex.tau.ac.il (U.N.); 3Israeli Rat Genome Center, Laboratory for Molecular Medicine, Barzilai University Medical Center, Ashkelon 78306, Israel; chyagil@gmail.com (C.Y.); labmomed@bgu.ac.il (Y.Y.); 4Faculty of Health Sciences, Ben-Gurion University of the Negev, Beer-Sheva 8410501, Israel; 5Department of Neurology, Rabin Medical Center, Petah-Tikva 49100, Israel; 6Sagol School of Neuroscience, Tel Aviv University, Tel Aviv 69978, Israel

**Keywords:** cerebral small vessel disease, white matter hyperintensity, stroke, hypertension, oxidative stress

## Abstract

Cerebral small vessel disease (CSVD) is the second most common cause of stroke and a major contributor to dementia. Manifestations of CSVD include cerebral microbleeds, intracerebral hemorrhages (ICH), lacunar infarcts, white matter hyperintensities (WMH) and enlarged perivascular spaces. Chronic hypertensive models have been found to reproduce most key features of the disease. Nevertheless, no animal models have been identified to reflect all different aspects of the human disease. Here, we described a novel model for CSVD using salt-sensitive ‘Sabra’ hypertension-prone rats (SBH/y), which display chronic hypertension and enhanced peripheral oxidative stress. SBH/y rats were either administered deoxycorticosteroid acetate (DOCA) (referred to as SBH/y-DOCA rats) or sham-operated and provided with 1% NaCl in drinking water. Rats underwent neurological assessment and behavioral testing, followed by ex vivo MRI and biochemical and histological analyses. SBH/y-DOCA rats show a neurological decline and cognitive impairment and present multiple cerebrovascular pathologies associated with CSVD, such as ICH, lacunes, enlarged perivascular spaces, blood vessel stenosis, BBB permeability and inflammation. Remarkably, SBH/y-DOCA rats show severe white matter pathology as well as WMH, which are rarely reported in commonly used models. Our model may serve as a novel platform for further understanding the mechanisms underlying CSVD and for testing novel therapeutics.

## 1. Introduction

Cerebral small vessel disease (CSVD) incorporates all disorders of the brain’s small perforating arterioles, capillaries and venules that cause lesions in the brain parenchyma [1]. Common manifestations include cerebral microbleeds, intracerebral hemorrhages, lacunar infarcts, white matter hyperintensities (WMH) and enlarged perivascular spaces [2]. CSVD accounts for 25% of ischemic strokes [1], more than 90% of intracerebral hemorrhages and 45% of cases of dementia in the elderly [3,4].

The etiology of CSVD is complex, and the mechanisms are type-dependent [5]. CSVD can be generally categorized into two main forms: the amyloidal form, which includes cerebral amyloid angiopathy (CAA), a chronic degenerative disease; and the non-amyloidal type, which is often related to common vascular risk factors, such as aging, diabetes mellitus and hypertension [6].

In clinical studies, 70–80% of CSVD patients are reported to be hypertensive [7]. Hypertension appears to be involved in both functional and structural alterations in blood vessels through multiple mechanisms [8]. Hypertension has been linked to the production of reactive oxygen species (ROS) and endothelial dysfunction [9], leading to blood–brain barrier (BBB) disruption [10]. In addition, when blood pressure is above the autoregulatory range, control of flow is lost and vasogenic edema occurs [9]. Finally, loss of vasoactive signals derived from neurons, astrocytes and the vessels themselves impair functional hyperemia; therefore, they prevent the ability to restore cerebral blood flow (CBF), ultimately leading to vasogenic edema [8].

Oxidative stress is not only a consequence of hypertension but also one of the fundamental mechanisms responsible for its development [11]. Physiologically, ROS controls vascular function by modulating various redox-sensitive signaling pathways [12]. In many studies, the endothelium was found to be the direct target of oxidative stress [13]. In response, endothelial cells become activated and produce vasoconstrictive agents [12]. Vascular oxidative stress promotes systemic inflammation via immune activation, and under conditions of oxidative imbalance, ROS generation stimulates atherogenesis [12,14]. Ultimately these processes lead to vascular disease, making oxidative stress a clear factor that strongly impacts the etiology of CSVD.

The pathogenesis of CSVD is not entirely clear, but it is likely that a variety of pathological mechanisms underlie the development of the disease. Although various experimental models were employed to study CSVD, chronic hypertensive models were found to resemble most key features of the disease, with spontaneously hypertensive stroke-prone rats (SHRSP) being the most-used model [15]. Nevertheless, no single animal model has been identified that reproduces all aspects of the human disease [16,17].

The Sabra model of salt-sensitive hypertension consists of two sub-strains, the Sabra hypertension prone (SBH/y) and hypertension resistant (SBN/y) rats. These sub-strains have been selectively inbred for over 90 generations for their susceptibility (sensitivity or resistance) to become hypertensive during salt loading. This model was extensively studied, and its cardiovascular and renal phenotypes were well characterized [18,19,20]. Interestingly, SBH/y rats were also shown to have an increased proportion of activated neutrophils and enhanced serum oxidative stress levels [21]. Due to the involvement of hypertension and oxidative stress in CSVD pathogenesis, we hypothesize that DOCA-treated SBH/y rats may serve as a novel model for CSVD and set out to investigate CNS vasculature involvement in this murine model of salt-sensitive hypertension.

## 2. Results

### 2.1. SBH/y Rats Develop High SBP and Neurological Deficits Following DOCA Administration

Hypertension was induced in SBH/y rats by DOCA and salt administration (‘SBH/y-DOCA’), as indicated by a substantial increase in SBP (Figure 1A), as compared to sham-operated SBH/y rats (‘SBH/y sham’). The effect of strain-specific effect of DOCA on SBP was verified by observing no elevation in SBP in SBN/y rats, which underwent the same procedure (‘SBN/y-DOCA’) as compared to sham-operated SBN/y rats (‘SBN/y-sham’). SBH/y-DOCA rats also showed significantly slower growth in body weight compared to SBH/y sham (Appendix A).

The neurological function of rats was assessed by serial neurological examinations (see Section 4). Starting from day 32 after induction of hypertension, SBH/y-DOCA rats showed a steady increase in neurological impairment, including weakness, lethargy, inability to walk straight, rotational behavior and stability deficiency (Figure 1B). Among the other groups, only minor neurological deficits were detected. By the end of the experiment, 60% of the rats in the SBH/y-DOCA group either died spontaneously or were sacrificed (Appendix A). Cortices from SBH/y-DOCA rats displayed multiple bleeding foci as well as dark regions interpreted as fluid-filled cavities, while SBH/y sham brains retained a normal appearance (Figure 1C).

In order to validate previously described induction in blood ROS levels in SBH/y-DOCA rats [21], plasma GSH levels were measured. Indeed, blood GSH levels in SBH/y-DOCA rats were significantly lower than those in SBH/y sham rats (Figure 1D). In addition, the levels of plasmatic cell-free dsDNA in SBH/y-DOCA rats were uniquely elevated, indicating induced NETosis (Figure 1E). Strikingly, plasmatic dsDNA levels strongly correlated with neurological scores, suggesting that NETosis levels may have a strong effect on clinical outcomes (Figure 1F).

### 2.2. SBH/y Rats on MRI

The pathological changes evident in the brains of SBH/y-DOCA rats were further characterized using MRI. Susceptibility weighted imaging (SWI), T1 and T2-weighted scans of SBH/y-DOCA cortices revealed evidence of stroke lesions such as ICH and the appearance of T2-hyperintense fluid-filled cavities, interpreted as lacunes reflecting previous infarcts (Figure 2A). These features were predominantly located in the posterior cerebral cortex. Corresponding histological analysis of the pathological areas revealed the presence of infiltrating erythrocytes, as well as spongy ischemic tissue, confirming the MRI findings (Figure 2B).

We hypothesized that brain pathology in the SBH/y-DOCA rats might include a change in volume, including cortical shrinkage and ventricular enlargement reflecting atrophy or cortical swelling due to edema. Therefore, we measured the volumes of these regions in SBH/y sham and SBH/y-DOCA rats. No significant differences were detected between sham and DOCA treated SBH/y rats in the volumes of cortices or ventricles (Appendix A). Interestingly, high variability in the ventricle volume of SBH/y-DOCA rats was observed compared to that of SBH/y sham (Appendix A). Average cortical T2 values of these groups were also not significantly different (Appendix A). However, within the SBH/y-DOCA group, we identified a strong correlation between cortical T2 values and the last neurological score, suggesting that cortical T2 values are indicative of disease severity (Figure 2C). Since high cortical T2 values may indicate higher brain water content as a result of edema, these results link edema and disease severity. Another indication of the potential contribution of edema to disease pathology is the significant association between increased cortical volume and the number of superficially damaged brain foci (Figure 2D).

### 2.3. Pathological Assessment of SBH/y-DOCA Brains

Histological assessment of brain slices derived from SBH/y sham and SBH/y-DOCA brains demonstrated multiple pathological features indicative of cerebrovascular disease in the brains of SBH/y-DOCA rats that were absent in the control group. These included ICH, ischemic regions, accumulation of erythrocytes and enlarged perivascular spaces (Figure 3A–H). Analysis of the distribution of ICH throughout the brain confirmed that most events were present in the cerebral cortex (Figure 3D), while an analysis of erythrocyte accumulation in these brain lobes indicated a more evenly dispersed distribution (Figure 3H).

Western blot analyses of PSD95 and GFAP were performed to quantify brain damage of synaptic plasticity and inflammation levels, respectively. The expression of PSD95 was significantly reduced in SBH/y-DOCA rat brains, indicating severe neuronal loss, while GFAP expression was sharply elevated in these brains, indicating enhanced astrocyte activation (Figure 3I,J).

### 2.4. Blood Vessel Pathology in SBH/y-DOCA Rats

Hypertension is known to alter cerebral pial arteries, including hypertrophic remodeling of the vessel wall. Thus, examination of wall-to-lumen ratio (WLR) on pial vessels of SBH/y-DOCA brains showed vessel hypertrophy as expected (Figure 4A). Moreover, the content of α-smooth muscle actin (αSMA) and rat-endothelial cell antibody (RECA-1) in underlying capillary beds, which perfuse the brain parenchyma, were assessed using immunostaining. As shown in Figure 4B, parenchymal blood vessel rarefaction as parenchymal blood vessels in SBH/y-DOCA brains had a packed morphology, were less dispersed throughout the brain and had smaller vessel areas than those of SBH/y sham brains. Such packed morphology may result in reduced CBF and capillary stenosis, ultimately leading to hypoperfusion. These findings may also explain the erythrocyte accumulation shown earlier (Figure 3D). Furthermore, increased immunostaining for the neutrophil effector protein myeloperoxidase (MPO) was observed in SBH/y-DOCA brain blood vessels, indicating that neutrophils contribute to disease pathology (Figure 4C).

### 2.5. BBB Permeability and White Matter Inflammation

We stained brains with rat IgG to estimate the extent of BBB permeability that may be a direct consequence of vessel damage. While mild IgG staining of surrounding brain blood vessels in sham-operated SBH/y rats was occasionally observed, robust and significantly elevated IgG immunostaining was found in the blood vessels of SBH/y-DOCA rats (Figure 4D). Surprisingly, IgG staining in these rats was not located proximal to blood vessels but localized to cortical layer VI, proximal to the corpus callosum (Figure 4D). Interestingly, GFAP immunostaining was also localized to the same region—indicating an elevated inflammatory response in the interface between the corpus callosum and layer VI of the cortex (Figure 4E).

### 2.6. White Matter Pathology in SBH/y-DOCA Brains

Luxol blue-cresyl violet staining of SBH/y sham and SBH/y-DOCA brains revealed their markedly different white matter structure. SBH/y-DOCA brains showed a thicker corpus callosum layer with an uneven luxol-blue staining pattern (Figure 5A and Appendix A). Interestingly, a nuclei-poor layer was apparent in the interface between layer VI of the cortex and the corpus callosum, which correlates with the area hyperstained with IgG and GFAP (middle panel in Figure 4D,E and Figure 5A).

T2 MRI analysis showed evidence of WMH in the corpus callosum of 4/9 of the SBH/y-DOCA brains analyzed, which was absent in control rats (Figure 5B, red arrow). Interestingly, a ‘halos’ shaped hyperintensity was identified in the interface between the corpus callosum and the cortex of SBH/y-DOCA rats, hereafter designated ‘interface hyperintensity’ (Figure 5B, green arrow and right panel). These interface hyperintensity halos were shown in the brains in which a poor nuclei layer was found by luxol blue-cresyl violet staining, indicating that they represent the same phenomena. Moreover, T2 values in this region significantly correlated with corpus callosum T2 values, indicating that interface hyperintensity is a manifestation of corpus callosum T2 hyperintensity (Figure 5D). Interestingly, while T2 values in the corpus callosum were not significantly different between the treatment groups (Appendix A), there was only a correlation between corpus callosum volume and T2 values for SBH/y-DOCA brains. Both of these indices can reflect white matter edema (Figure 5E). Furthermore, Western blot analysis indicates a significant reduction in MBP protein levels in SBH/y-DOCA rat brains, further illustrating the white matter pathology in these rats (Figure 5F).

### 2.7. Cognitive Decline in SBH/y DOCA Rats

We performed behavioral examinations to test whether the above-mentioned pathological changes correlated with cognitive and memory deficits. We subjected SBH/y sham and SBH/y-DOCA rats to the novel object recognition (NOR) test, measuring investigative behavior together with learning capacity and memory. In the training phase, rats were familiarized with two similar objects. Subsequently, one object was substituted with a novel object in the test trial 24 h later (Figure 6A). SBH/y-DOCA rats spent less time sniffing the novel object than SBH/y sham rats, as shown by heatmaps, preference index and total exploration time, indicating a reduction in recognition memory (Figure 6B–D). The shorter distance SBH/y-DOCA rats traveled throughout the training arena indicated a less exploratory behavior (Figure 6E). This finding indicates that a lower preference of SBH/y-DOCA rats towards the novel object may reflect memory deficits combined with less exploratory behavior. Hence, to specifically measure associative learning and memory deficits, we performed a contextual fear conditioning test, in which rats were conditioned to the arena with a hostile shock, followed by measuring their freezing behavior 24 h later. Indeed, memory deficits were apparent for SBH/y-DOCA rats, as indicated by the shorter freezing time (Figure 6F). Together, these analyses clearly demonstrate robust cognitive deficits involving reduced exploratory behavior and memory impairment in SBH/y-DOCA rats.

## 3. Discussion

Small vessel arteriopathy is a chronic disease that progresses over the course of years. In the current study, we attempted to generate an accelerated model that would simulate CSVD by exposing the animal’s brain vasculature to hypertension, a major risk factor for CSVD. We examined cerebrovascular pathology in SBH/y rats showing hypersensitivity for blood pressure elevation upon DOCA administration and enhanced peripheral oxidative stress mediated by neutrophils. We found that SBH/y-DOCA rats display neurological and cognitive deficits, multiple ICH and lacunes, WMHs, evidence for blood vessel occlusions, enlarged perivascular spaces and evidence of vasogenic edema. Notably, there is a strong resemblance to human SVD, as seen in the white matter changes, parenchymal microbleeds and clinically cognitive decline and motor deficits.

CSVD plays a crucial role in cerebrovascular disease and leads to stroke and vascular dementia. However, human sporadic CSVD is classified solely based on imaging patterns, with no peripheral biomarkers available for clinical use [6]. The identification of such biomarkers, as well as gaining insight regarding disease ontogeny, is further hampered by the fact that CSVD is accumulative and often underdiagnosed at the early stages [1], leading to improper risk factors management and rapid disease progression. Thus, rodent models of CSVD might be used as excellent tools for gaining a deeper understanding of the different mechanisms of action that result in CSVD, as well as for discovering novel diagnostic tools and therapeutics [17].

Current models of CSVD, such as the SHRSP rats and NOTCH3 transgenic mouse model of CADASIL, were instructive regarding CSVD pathology [17]. However, both SHRSP rats and NOTCH3 transgenic mice reproduce only specific aspects of CSVD, with severe endothelial injury and BBB pathology in SHRSP and apparent white matter pathology in NOTCH3 transgenic mice [17]. Since the pathologic mechanisms of CSVD remain elusive, there is a need for additional models encompassing specific pathogenic events, such as WMH and enlarged perivascular spaces [15,17].

One possible strategy for better disease modeling is concomitantly addressing several pathogenic events known to contribute to the human disease. In this context, one prominent pathogenic event in CSVD is peripheral oxidative damage [13,17]. Therefore, we speculated that combining hypertension and peripheral oxidative stress can enable improved modeling of cerebral blood vessel stress and associated brain damage, considering the pathogenic role of neutrophils in cerebral brain vessels. Indeed, we witnessed severe blood vessel pathology characterized by hypertrophy in pial vessels, stenosis and deterioration of cerebral capillaries. Profound BBB pathology was also recorded with the accumulation of blood-borne proteins in the interface of the corpus callosum and cortical layer VI.

One novel observation in our findings is the occurrence of WMH in SBH/y-DOCA rat corpus callosum. While diffuse white matter damage is often seen in hypoperfusion-based models [15], WMH is rarely reported in other experimental rodents, with conflicting results in the SHRSP model [22,23]. In addition to the patchy, periventricular WMH seen in our model, we also witnessed other white matter abnormalities such as myelin loss, thickened corpus callosum and, most intriguingly, T2 hyperintensity in the interface between the white matter and grey matter. 

Based on the correlations between interface T2 values, corpus callosum T2 values and corpus callosum volume, we interpreted interface hyperintensities as fluid accumulation, probably resulting in an in situ cellular toxicity due to the presence of toxic blood-borne proteins and leading to inflammation and neuronal loss.

Human WMH is heterogeneous and is hypothesized to result from either hypoperfusion of a brain region, resulting in local ischemia, or from non-ischemic events. Local ischemia forms a fluid-filled lacune, while non-ischemic events, such as fluid leakage from the ventricles into the white matter due to disruption of the ependymal cell layer lining the ventricles, result in periventricular WMH [24]. Such periventricular WMH is often seen in the early stages of disease together with subependymal gliosis, reduced myelin staining and edema [25]. The simultaneous occurrence of these findings in our model may imply that interface hyperintensity results from non-ischemic white matter damages, perhaps due to ependymal layer breach. However, further analysis of the fluid route in this model is needed to confirm this hypothesis.

Cerebral microbleeds identified in SBH/y-DOCA rats were predominantly localized to perforating arterioles in the cerebral cortex. In human CSVD, such bleeding often indicates an amyloid-beta pathology in the vessel wall, resulting in CAA. However, we were not able to detect amyloid deposition in SBH/y-DOCA brain vessel walls using Congo Red staining nor through immunohistochemistry. Amyloid-beta pathology following hypertension was detected previously in SHRSP rats, in senescent rats (30 weeks or more) and after CSVD-like pathology accumulation, indicating that amyloid pathology was only secondary to hypertension [26]. However, the question of why bleeding occurred predominantly in the cortex remains unclear. In humans, cerebral microbleeds due to hypertension are localized to deep brain regions, as the pulse pressure is the highest among these large arteries [27]. The altered distribution of pulse pressure throughout brain vessels in rats may explain the change in microbleed localization observed in our model. This possibility is further supported by work with SHRSP rats showing that ~30% of the bleedings in adult rats were localized to the cerebral cortex [28]. Furthermore, it was shown that rat pial arterioles blood vessels are exposed to increased pulse pressures, similar to sub-cortical areas in the human brain, which may trigger vessel hypertrophy and loss of autoregulation [29].

NETosis levels were elevated in SBH/y-DOCA rats as compared to sham-treated rats, and there was a strong correlation between NETosis levels and neurological scores. Peripheral oxidative stress levels were also increased in SBH/y-DOCA rats, and MPO accumulated in parenchymal brain vessels. Importantly, the results demonstrate an association between NETosis rates and neurological deficits in our model. This association is not surprising considering the recently identified role of NETosis in murine models of stroke, where infiltrating neutrophils were shown to form NETosis in the brain parenchyma [30] and in stroke patients, where NET levels were significantly associated with mortality rates [31] and elevated in injured brain areas [32]. One prominent hypothesis is that NETosis has a role in thrombosis formation, as extracellularly released dsDNA serves as a scaffold for cells and coagulation factors. In this context, high motility group box 1 (HMGB1), a DNA-binding protein known to be associated with cerebrovascular disease [33,34], was suggested to play a critical role in inducing NETosis [35]. However, the robustness of this association implies that NETosis might be more dominant in this cascade than previously assumed.

CSVD is yet an incurable disease. SBH/y-DOCA rats may serve as a novel platform for testing potential therapeutics, such as the FDA-approved drug trifluoperazine (TFP). TFP, a calmodulin inhibitor, was effectively shown to reduce cerebral edema by inhibiting aquaporin-4 (AQP4) localization [36,37]. Although the detailed molecular mechanism of brain water transport is lacking, AQP4 was implicated as a key determinant in the lymphatic system and, therefore, a potential therapeutic target for brain disorders presenting edema [38,39,40,41,42]. As mentioned, CSVD is characterized by various manifestations that can be utilized as therapeutic candidates to be evaluated using evolving approaches for drug discovery and development. High throughput screening (HTS) and computer-aided drug design (CADD) are examples of such approaches, gaining interest in recent years [43,44]. The importance of these tools lies in the improved drug screening efficiency with higher predictability and clinical applicability. Nevertheless, reliable animal models, such as SBH/y-DOCA rats, are still required for testing target validation and functional efficiency.

In summary, we identified and characterized a novel model for CSVD that combines chronic hypertension with peripheral oxidative stress. Additional studies in this model should broaden our understanding of specific pathologic events associated with CSVD and provide a platform for testing novel therapeutics.

## 4. Materials and Methods

### 4.1. Animal Housing

Male Sabra hypertension-prone rats (SBH/y) and Sabra hypertension-resistant rats (SBN/y) were obtained from the Israeli Rat Genome Center at the Barzilai University Medical Center (Ashkelon, Israel). Rats were maintained in 12 h light/12 h dark conditions in acclimatized rooms and had free access to food and water. All experimental protocols were authorized by the Tel Aviv University Committee of Animal Use for Research and Education (protocol code 01-20-026, 11.5.2020). Every effort was made to reduce the number of rats used and minimize their suffering.

### 4.2. Experimental Design and Induction of Hypertension

Animals were weaned at age six weeks and randomly assigned to experimental or control groups. Isoflurane was used for both induction (5%) and maintenance (1.5–2%) of anesthesia. Salt-loaded animals (‘SBN/y-DOCA’ (*n* = 4) and ‘SBH/y-DOCA’ (*n* = 19) groups) were subcutaneously implanted with 75 mg deoxycorticosterone-acetate (DOCA) pellets (Innovative Research of America) and provided with 1% NaCl and 0.001% Potassium Chloride in tap water. Sham control animals (‘SBN/y sham’ (*n* = 4) and ‘SBH/y sham’ (*n* = 12) groups) were anesthetized, a skin incision was made, and the incision was sewn closed. Control animals were provided with tap water with no added salt. Animals were followed daily for a period of two months. When severe symptoms of stroke or loss of >10% body weight/week occurred, or it appeared that death was imminent, animals were sacrificed. Sacrificed animals were counted in survival curves, and tissues were harvested and used for analyses.

### 4.3. Systolic Blood Pressure Measurements

Systolic blood pressure (SBP) was measured in conscious rats using the CODA™ non-invasive tail-cuff system (Kent Scientific, Torrington, CT, USA). Acclimation to the restrainer chamber was performed for 5 days. On the first two days, rats were inured to the chamber on a heating platform for 5 min. On the third and fourth days, tail cuffs were added and SBP was measured. On the fifth day, baseline measurements were taken. Subsequent measurements were recorded on days 14, 21, 28 and 42. Each measurement session consisted of 12 consecutive measurements. Criteria for data acceptance were the acquisition of at least five measurements and SD < 30 for each session.

### 4.4. Neurological Deficits Score

A neurological assessment method was adapted and modified from Hunter et al. [45]. The assessment comprised serial examinations, including assessment of paw placement and flexion, ability to grip and stabilize on a horizontal bar, visual forepaw reaching, contralateral rotation, Pinna and corneal reflexes, general condition, motility, limpness and circling. Scoring was performed twice a week from day 30 until termination. For animals sacrificed due to ethics guidelines, clinical scores were fixed as the last score prior to sacrifice.

### 4.5. Behavioral Examinations

For acclimation, animals were transferred to a dedicated behavioral testing room 30 min prior to the beginning of each trial. Chambers were cleaned with Virusolve+ between trials. Tests were monitored and analyzed using an automated tracking system (Ethovision, Noldus).

### 4.6. Novel Object Recognition (NOR) Test

The test was conducted in an open plastic chamber (50 × 50 × 50 cm) with white walls and a black floor. One week prior to testing, rats were allowed to explore the box for 10 min for acclimation. During the training session, on experimental day 49, two identical objects were placed in the chamber, and rats explored the chamber for 10 min. Twenty-four hours later, one of the familiar objects was substituted with a novel object, and rats were allowed to explore the chamber again for five minutes. Evaluation of NOR memory was expressed as a percentage of the preference ratio that was calculated according to the following formula: Preference index = N/(N + F), where N represents the time spent sniffing the new object and F represents the time spent in sniffing the familiar object. Total exploration time and the distance traveled throughout the training session were also calculated.

### 4.7. Contextual Fear Conditioning

The test was performed in a fear conditioning chamber (17 × 17 × 25 cm) on an experimental day 41. On day 1, animals were placed in the chamber for 3 min for acclimation. On day 2, animals were again placed in the chamber for 5 min. After 2 min, the rats received two foot shocks (0.7 mA, 2 s) at a two-minute interval. On day 3, the rats were placed in the chamber for 3 min. Fear conditioning was evaluated by scoring freezing behavior on the third day, i.e., the absence of all movement except for respiration.

### 4.8. Tissue Collection and Sample Preparation

For immunostaining and MRI, rats were anesthetized with ketamine/xylazine and transcardially perfused with 50 mL phosphate-buffered saline (PBS) followed by 200 mL of 4% paraformaldehyde in PBS (PFA-PBS). Brains were removed and fixed for 24 h in PFA-PBS prior to MRI. Brains were transferred to PBS for 12 h prior to imaging and were transferred again into an NMR tube in Fluorinert FC-40 solution (Sigma-Aldrich, St. Louis, MO, USA) immediately before imaging. Brains were fixed for 48 h in PFA-PBS and then placed in 30% sucrose solution for another 48 h prior to immunostaining. Brains were stored at 4 °C in PBS supplemented with 0.01% sodium azide until sections were prepared. The brains were cut into hemispheres, embedded in OCT and frozen on dry ice. Coronal sections (20 μm) were cut using a freezing sliding microtome (Leica CM1850, Wetzlar, Germany) and stored at −20 °C until use. Rats were anesthetized with CO_2_ and decapitated prior to preparing samples for protein analysis. The brains were immediately removed, and cortices were snap-frozen in liquid nitrogen and stored at –80 °C.

### 4.9. Protein Extraction and Western Blot

Tissues were thawed and immediately mechanically ground in RIPA buffer (Thermo Fisher Scientific, Waltham, MA, USA), freshly added with protease and phosphatase inhibitors (1:100, Thermo Fisher Scientific), incubated 30 min on ice, and centrifuged at 14,000 RPM for 20 min at 4 °C. Protein levels were determined using a BCA kit (Thermo Fisher Scientific). Supernatants were stored at −80 °C until used. An amount of 30 µg proteins were separated using electrophoresis through 4–20% polyacrylamide gradient gels (Thermo Fisher Scientific). Following electrophoresis, the protein was transferred to nitrocellulose membranes using transblot equipment. The nitrocellulose membranes containing protein samples, which were transferred from polyacrylamide gels, were blocked using super block (Thermo Fisher Scientific) with 0.1% Tween 20 (Thermo Fisher Scientific). The membranes were probed overnight at 4 °C with the primary antibodies: GFAP (1:6500, ab7260, Abcam, Cambridge, MA, USA), MBP (1:1000, ab7349, Abcam) and PSD-95 (1:1000, ab18258, Abcam). Membranes were washed three times with Tris Buffer Saline 0.1% Tween 20 (TBST) and incubated with secondary antibodies: goat anti-mouse or goat anti-rabbit IRDye^®^ 800CW/680CW (1:10,000, Licor, Lincoln, NE, USA) for one hour at room temperature (RT). The membranes were then developed with Odyssey Imager CLx (Licor). As a control for protein loading, blots were subsequently probed for mouse anti-GAPDH (1:1000, MAB #2118, Cell-Signaling, Danvers, MA, USA) using the same procedures. Data were calculated as the ratio of mean target protein intensity to GAPDH intensity. Densitometric analysis of Western blots was performed using Odyssey 2.1 software (Licor).

### 4.10. Histological Stainings

Three slices per brain were stained with H&E-DAB or cresyl violet and luxol fast blue. Briefly, thawed slides were washed twice with PBS and either incubated for five minutes with DAB (Thermo-Fisher) according to the manufacturer’s protocol, then stained with H&E (Sigma-Aldrich) according to standard protocols or stained with Luxol Fast Blue (Sigma-Aldrich, St. Louis, MO, USA) followed by cresyl violet (Sigma-Aldrich) according to standard protocols. H&E-DAB staining was used to evaluate cerebral edema, ICH, immune cell infiltration, lacunar infarcts, perivascular spaces and red blood cell (RBC) accumulation. Cresyl violet and luxol staining were used to evaluate white matter size as well as the wall-to-lumen ratio (WLR) of pial vessels.

### 4.11. Immunohistochemistry

Slides were thawed and washed twice with PBS; blocked and permeabilized with PBS 1% bovine serum albumin, 5% goat serum (Biological Industries, Beit Ha’emek, Israel) and 0.05% Triton-X (Sigma-Aldrich) for one hour; and incubated with the following primary antibodies overnight at 4 °C: αSMA (1/400, 19245, Cell Signaling, Danvers, MA, USA), GFAP (1/1000, ab7260, Abcam), MPO (1/100, ab90810, Abcam), RECA (1/100, ab9774, Abcam). Slides were washed three times with PBS and incubated with a fluorescent-labeled secondary antibody (1/700, Alexa-Flour) for 1 h at RT. For IgG staining, brains were incubated overnight with 1/200 Alexa-Flour Cy3 anti-rat IgG antibody. DNA was stained for 10 min with DAPI (1:1000, Sigma-Aldrich). Slides were mounted with Flouromount-G. αSMA and RECA images were used to calculate vessel area. In order to minimize the background signal, slides were incubated with secondary antibodies and without primary antibodies. This ensures that staining is produced from the detection of the antigen by the primary antibody and not by the detection system or the specimen.

### 4.12. Microscopy and Image Analysis

Histologically or immunohistochemistry-stained slides were either scanned at a magnification of x20 using the Aperio Versa slide scanner (Leica, NA = 0.8) or imaged using the SP8 confocal microscope (Leica, NA = 0.75). The vessel’s characteristics were calculated using the Aperio ImageScope software (Leica). Fluorescent intensity and % area stained were calculated using the ImageJ software. WLR of pial vessels was calculated by the formula: 0.5× [vessel diameter−lumen diameter]/lumen diameter. All image analyses were conducted blindly. Analyses parameters in ImageJ were maintained constant for every staining in all the slides in order to avoid biases and exclude the interference of background signals.

### 4.13. NETosis Levels Measurement

Blood samples were collected into EDTA tubes. Plasma was collected following centrifugation at 2000× *g* for 10 min in RT. Plasma levels of cell-free DNA were measured using a fluorometric assay for double-stranded DNA (dsDNA), Quant-iT PicoGreen (Invitrogen, Carlsbad, CA, USA) according to the manufacturer’s protocol.

### 4.14. Glutathione (GSH) Levels Measurement

Blood GSH levels were measured using the GSH-GLO Glutathione Assay (Promega, Madison, WI, USA) and according to the manufacturer’s instructions.

### 4.15. MRI

#### 4.15.1. Imaging Protocols

All scans were performed ex vivo on a vertical bore Bruker Ascend 9.4T WB NMR spectrometer (Bruker Biospin, Billerica, MA, USA), using a Bruker Micro 5 probe capable of producing pulsed-field gradients of up to 300 G/cm.

T2w scans: Samples were scanned using a multi-slice–multi-echo (MSME) sequence, with TE/TR= 15/6000 ms, 20 echoes and 4 averages. Twenty slices were acquired with 256 × 256 pixels matrix, yielding a resolution of 0.039 × 0.039 × 0.5 mm.

T1w scans: A multi-TR sequence was used, with TE = 7.6 ms and 7 TR values ranging exponentially between 400 and 6000 ms with 4 averages. The image matrix and resolution were the same as mentioned above.

Susceptibility Imaging (SWI): Images were acquired using a fast-low-angle shot (FLASH) sequence, with a flip angle of 30 deg, TE/TR = 45/1090 ms and 8 averages [46,47]. The image matrix and resolution were the same as mentioned above.

#### 4.15.2. MRI Data Analysis

T_2_ and T_1_ weighted images were analyzed based on Equations (1) and (2) using nonlinear least-squares in MATLAB (Mathworks, Natick, MA, USA) for the calculation of T_2_ and T_1_ maps, respectively.
(1)$$S\left(t\right)=S_{0}\mathrm{expexp}\;\left(-\frac{t}{T_{2}}\right)$$
(2)$$\left(t\right)=S_{0}\left(-\frac{t}{T_{1}}\right)$$

The values of T_2_ and T_1_ were extracted only for voxels exhibiting a good fit (R^2^ > 0.95).

Additionally, regions of interest (ROIs) for each animal and each slice were manually segmented based on a reference image: corpus callosum (CC), cortex, lateral ventricle and white matter and gray matter interface (Examples of segmentations are shown in Appendix A). The signal in the segmented data was summed and fitted to Equations (1) and (2) again to obtain an improved fitting (R^2^ > 0.999) for the parameters.

SWI was calculated based on the scheme described by Hsieh et al. [48] using the multiplication of the magnitude image and weighted filtered phase image in 4th power.

### 4.16. Statistical Analyses

All statistical analyses were performed using GraphPad Prism 7. Data were tested for normality using Kolmogorov−Smirnov and Shapiro−Wilk tests. Data are expressed as mean ± SD. SBP and neurological scores were analyzed using a Two-Way ANOVA followed by Tukey’s post hoc analyses. All other analyses were performed using an unpaired Student’s *t*-test. Correlation analyses were performed using Pearson Correlation. For all tests, the statistical significance threshold was set to *p <* 0.05.

## Figures and Tables

**Figure 1 ijms-23-05915-f001:**
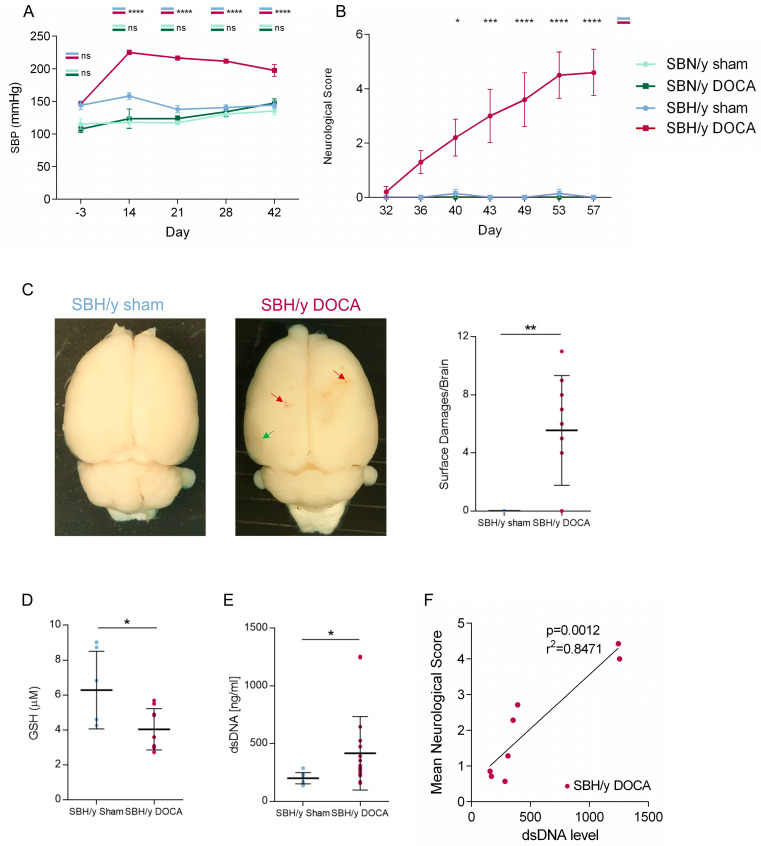
Hypertension, oxidative stress and neurological deficits in SBH/y-DOCA rats. Systolic blood pressure (**A**) and neurological scores (**B**) of SBH/y sham (*n* = 7), SBH/y-DOCA (*n* = 10), SBN/y sham (*n* = 4) and SBN/y-DOCA (*n* = 4) rats. Representative images and quantification (**C**) of brains derived from SBH/y sham (*n* = 7) and SBH/y-DOCA (*n* = 9) rats, showing superficial bleedings (red arrows) and dark ischemic regions (green arrow). Blood GSH levels in SBH/y sham (*n* = 6) and SBH/y-DOCA (*n* = 9) rats (**D**). dsDNA levels in plasma of SBH/y sham (*n* = 11) and SBH/y-DOCA (*n* = 19) rats (**E**) and a correlation of plasmatic dsDNA levels and mean neurological score (day 32 to day 57) in SBH/y-DOCA rats (*n* = 8) (**F**). Data are presented as mean ± SD. * *p* < 0.05, ** *p* < 0.01, *** *p* < 0.001, **** *p* < 0.0001. Two-way ANOVA analyses in (**A**,**B**). Two-tailed *t*-test analysis in **(C**–**E**). Pearson’s Correlation in (**F**).

**Figure 2 ijms-23-05915-f002:**
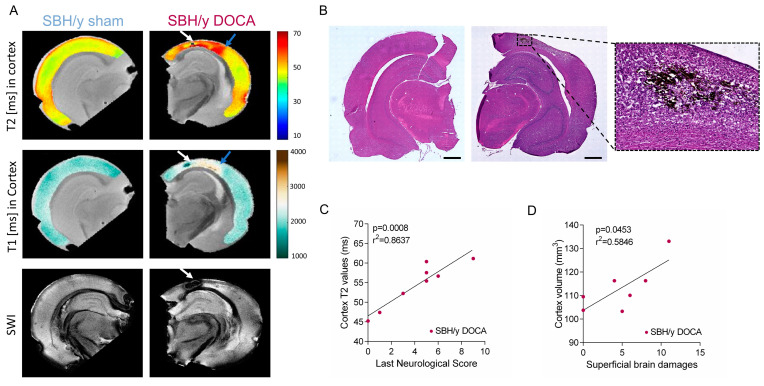
MRI correlates brain pathology in SBH/y-DOCA rats. Representative T2-weighted (top panel), T1-weighted (middle panel) and SWI (bottom panel) images of SBH/y sham (*n* = 5) and SBH/y-DOCA (*n* = 9) rats (**A**). White arrows indicate ICH, and blue arrows indicate fluid-filled cavities. Correlative H&E-DAB staining of brains is shown in (**A**) and 10× magnification of the injured area (**B**). Correlation between cortical T2 values and last neurological score in SBH/y-DOCA rats (*n* = 9) (**C**). Correlation between cortical volume and the amount of obvious brain damage in SBH/y-DOCA rats (*n* = 9) (**D**). Scale bar = 1 mm. Pearson correlation in (**C**,**D**).

**Figure 3 ijms-23-05915-f003:**
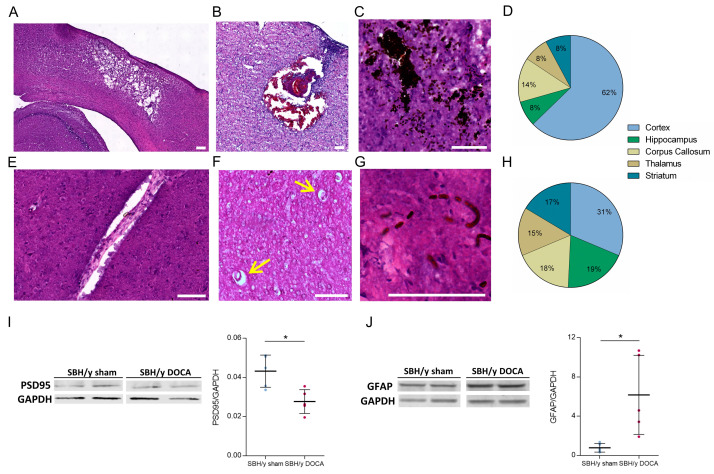
Pathological assessment of SBH/y-DOCA brains. H&E (**A**,**B**,**E**,**F**) and H&E-DAB (**C**,**G**) images of representative pathologies seen in brains of SBH/y-DOCA rats: Fluid-filled cavities surrounded by a spongy tissue (**A**,**B**), occluded vessels (**B**), ICHs (**C**), enlarged perivascular spaces (**E**,**F**, yellow arrows) and erythrocytes accumulating in capillaries (**G**). Distribution of the occurrence of ICHs and erythrocytes accumulations throughout the brain is shown in (**D**,**H**), respectively. PSD95 and GFAP Western blots of protein extracted from cerebral cortices of SBH/y sham (*n* = 5) and SBH/y-DOCA (*n* = 5) rats are shown in (**I**,**J**), respectively. Scalebars = 100 µm. Data are presented as mean ± SD, * *p* < 0.05, Two-tailed *t*-test.

**Figure 4 ijms-23-05915-f004:**
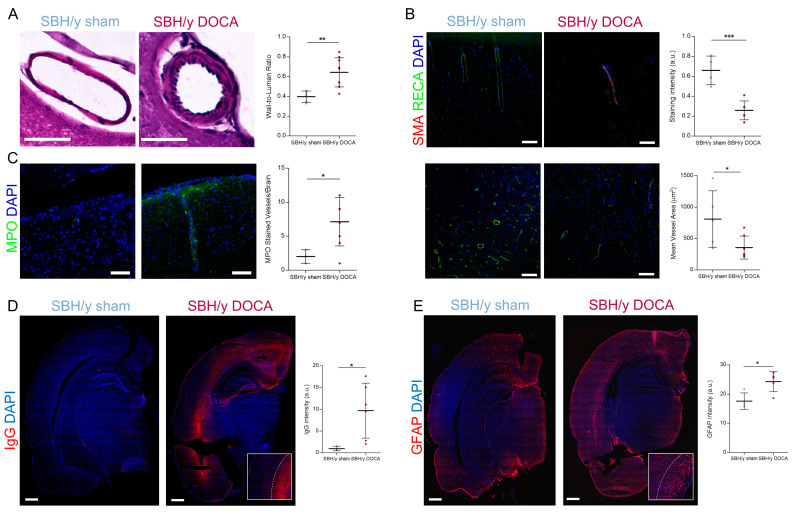
Vascular pathology and BBB permeability in SBH/y-DOCA brains. Representative H&E images and wall-to-lumen ration (WLR) quantification (**A**) of pial blood vessels of SBH/y sham (*n* = 5) and SBH/y-DOCA (*n*
*=* 9) rats. αSMA and RECA-1 co-immunostaining of parenchymal blood vessels (**B**) and quantification of mean staining intensity (top right) and mean vessel area (bottom right) in SBH/y sham (*n*
*=* 4) and SBH/y-DOCA (*n*
*=* 7) brains. Representative images and quantification (**C**) of parenchymal blood vessels staining for MPO in the brains of SBH/y sham (left, *n*
*=* 4) and SBH/y-DOCA (right, *n*
*=* 6) rats. IgG staining and quantification (**D**) and GFAP immunostaining and quantification (**E**) of SBH/y sham (*n*
*=* 4) and SBH/y-DOCA (*n* = 6) brains. Insets in (**D**,**E**) show a magnification outlining the corpus callosum border. Scale bars in (**A**–**C**) = 75 µm. Scale bars in (**D**,**E**) = 3 mm. Data are presented as mean ± SD. * *p* < 0.05, ** *p* < 0.01, *** *p* < 0.001, two-tailed *t*-test.

**Figure 5 ijms-23-05915-f005:**
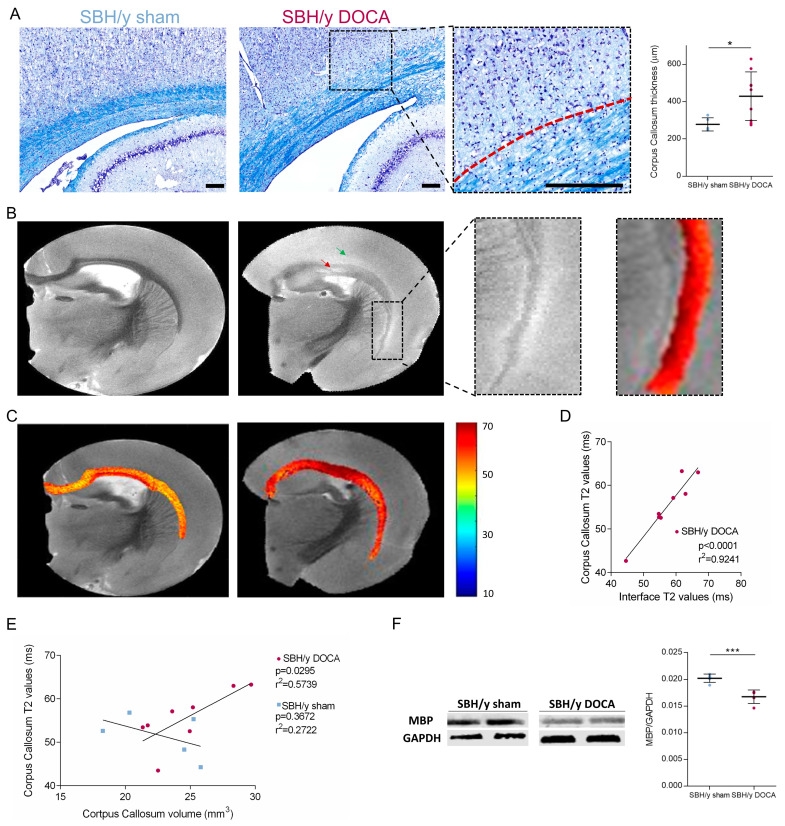
White matter pathology in SBH/y-DOCA brains. Representative luxol blue and cresyl violet staining of SBH/y sham and SBH/y-DOCA brains (**A**), and 5x magnification of the white matter–grey matter interface of SBH/y-DOCA (middle panel). Spotted red line in (middle panel in (**A**)) identifies the border between the white matter and the grey matter. Quantification of corpus callosum thickness in SBH/y sham (*n* = 5) and SBH/y-DOCA (*n* = 9) brains (right panel). T2 images of SBH/y sham and SBH/y-DOCA (**B**). Red arrow on (**B**) marks WMH, and green arrow marks an ‘interface hyperintensity halo’. 4× magnification of the white matter–grey matter interface of SBH/y-DOCA (middle panel in (**B**)) and T2 values of the same region (right panel in (**B**)). T2 values of corpus callosum of representative SBH/y sham and SBH/y DOCA brains (**C**). Correlation between T2 values in the white matter–grey matter interface and in the corpus callosum in SBH/y-DOCA rats (*n* = 9) (**D**). Correlation between corpus callosum volume and T2 values in SBH/y sham and SBH/y-DOCA brains (**E**). Representative Western blot images and quantification (**F**) of MBP in of SBH/y sham (*n* = 5) and SBH/y-DOCA (*n* = 5) cortical extracts. Scale bars = 200 µm. Data are presented as mean ± SD. * *p* < 0.05, *** *p* < 0.001. Two-tailed *t*-test analyses in (**A**,**F**). Pearson correlation in (**D**,**E**).

**Figure 6 ijms-23-05915-f006:**
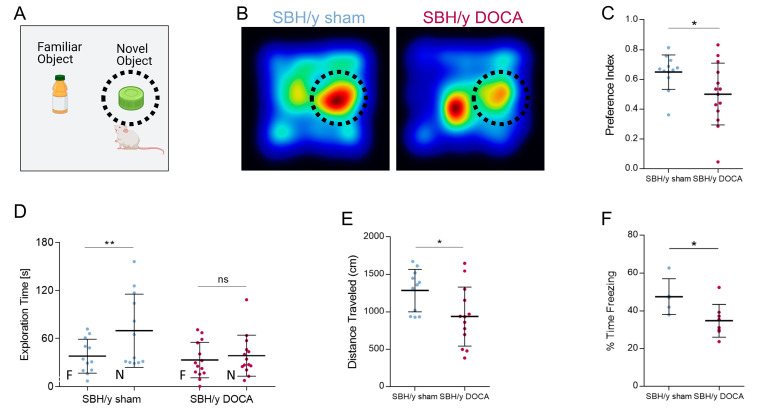
Cognitive deficits in SBH/y-DOCA rats. Novel object recognition (NOR) test schematic illustration (**A**) and heatmaps of averaged cumulative time spent next to the novel object (marked by dashed circle) of SBH/y sham (*n* = 12) and SBH/y-DOCA (*n* = 14) rats during the NOR test trial, 50 days following DOCA treatment (**B**). Preference index (**C**) and exploration time (**D**) of SBH/y sham and SBH/y-DOCA during the NOR test trial. Distance traveled during the NOR training trial **(E)**. Freezing time of SBH/y sham (*n* = 5) and SBH/y-DOCA (*n* = 8) rats 41 days following DOCA treatment during contextual fear conditioning test (**F**). Data are presented as mean ± SD. ** p* < 0.05, *** p* < 0.01, two-tailed *t*-test.

## Data Availability

The datasets generated and analyzed during the current study are available from the corresponding author on reasonable request.

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
