# Peer review of "A Novel Rodent Model of Hypertensive Cerebral Small Vessel Disease with White Matter Hyperintensities and Peripheral Oxidative Stress"

_ijms, 2022, doi:10.3390/ijms23115915_

Round 1

Reviewer 1 Report

The authors performed an elegant and extensive study based on “Sabra” hypertension-prone rats. They showed that this type of rats may produce a novel model of cerebral small vessel disease, closely resembling cerebral small vessel disease in humans. Studies in this model may improve our understanding of the biochemical mechanisms on this disorder, may help to reveal biomarkers useful for in vivo diagnosis and offer a model for testing therapeutic interventions.

The present study is well conducted, has novelty, the methods are more than adequate and it is very interesting. Furthermore, the Introduction and the Discussion section additionally offer a short but comprehensive and educative review of cerebral small vessel disease. The article deserves to be published.

One minor point. I would prefer the authors to add the term “hypertensive” in the title: Sabra rats as a novel model of hypertensive cerebral small vessel disease with white matter hyperintensities and peripheral oxidative stress.

Author Response

The authors would like to thank the reviewer.

The title was changed as suggested. Attached is the revised manuscript, including all changes following all of the reviewers comments.

Reviewer 2 Report

Dear Editor,

The manuscript by Guy et al. identified and characterized a novel model for CSVD that combines chronic hypertension with peripheral oxidative stress.

The design of the study and the technical quality of the work look convincing and results can be of general interest. The manuscript is well-written and easy to follow. However, there is a number of major and minor points that would need to be addressed in order to improve the quality of this paper before it can be accepted for publication:

Major:

-Imaging was an essential aspect of this manuscript. Author needs to provide more details such as how many FOVs have been taken and what are their measures to minimize biases, and how they have excluded any possible interference from background signals in order to enhance the reproducibility of the presented data. Magnification number should be included for all the figures. But it won’t be enough as it has nothing to do with resolution especially for the purpose of quantitative analyses like in this study. So, author needs to include NA of the utilized lens as well.

-Edema is a major hallmark for CSVD as the authors have indicated “We hypothesized that brain pathology in the SBH/y-DOCA rats may include change in volume, including cortical shrinkage and ventricular enlargement reflecting atrophy, or cortical swelling due to edema”. However, the discussion omits a breakthrough study from 2020 by Kitchen et al in Cell, regarding new paradigm to target edema via acting on BBB/BSCB and astrocytes. That study shows pharmacological inhibition of these signalling events prevents the development of CNS edema and promotes functional recovery in injured rats. This role has been recently been confirmed by the work of Sylvain et al BBA 2021 which has demonstrated that targeting astrocytes is a viable therapeutic target using a photothrombotic stroke model. They have also shown a link to brain energy metabolism as indicated by the increase of glycogen levels. References to be included:

https://pubmed.ncbi.nlm.nih.gov/32413299/

https://pubmed.ncbi.nlm.nih.gov/33561476/

This could provide a new insight to further explain this interesting finding by the authors where they stated “Surprisingly, IgG staining in these rats was not located proximal to blood vessels 192 but localized to cortical layer VI, proximal to the corpus callosum (Fig. 4D). Interestingly, 193 GFAP immunostaining was also localized to the same region – indicating an elevated in- 194 flammatory response in the interface between the corpus callosum and layer VI of the 195 cortex (Fig. 4E)”.

Minor:

-Authors need to discuss the emerging role of glymphatic pathway which plays an important role in dementia. It is a waste clearance system that utilizes a unique system of perivascular channels, to promote efficient elimination of soluble proteins and metabolites from the central nervous system. Developing new drugs against this system using this novel model will have huge therapeutic potential. References:

https://academic.oup.com/brain/advance-article/doi/10.1093/brain/awab311/6367770

https://www.nature.com/articles/s41583-021-00514-z

Following recent trends in targeting the molecular and signalling mechanisms involved in edema and hypertension rather than just the traditional approaches using this new model will be very interesting. The importance of this new approach has been discussed in these references which should be included to enrich the discussion of current manuscript. References:

https://pubmed.ncbi.nlm.nih.gov/34973181/

https://pubmed.ncbi.nlm.nih.gov/34863533/

https://pubmed.ncbi.nlm.nih.gov/35163313/

-End of discussion and towards the conclusion: CSVD is yet an incurable disease. Author needs to point out to the recent advances in applying the use of high-throughput screening and computer-aided drug design as have been nicely reviewed by Aldewachi et al 2021 as they can provide a novel insight that can support target validation in future studies. References to be included:

https://pubmed.ncbi.nlm.nih.gov/33925236/

https://pubmed.ncbi.nlm.nih.gov/33672148/

Best.

Author Response

The authors would like to thank the reviewers for reading and evaluating our paper. We have addressed the major comments and revised the manuscript to the best of our abilities. Below can be found detailed reference to each comment from the reviewers.

First comment:

Three FOVs from each animal were taken randomly and all image analyses were done blindly in order to minimize biases. In order to minimize background signal, slides were incubated with secondary antibody and without primary antibody. This ensure that staining is produced from detection of the antigen by the primary antibody and not by the detection system or the specimen. In addition, all microscope and analysis parameters in ImageJ were maintained constant for every staining in all the slides in order to avoid biases and exclude interference of background signals.

As mentioned in the methods section, histologically or immunohistochemistry-stained slides were either scanned at magnification of x20 using the Aperio Versa slide scanner (Leica) or imaged using the SP8 confocal microscope (Leica). For Aperio Versa slide scanner – NA is 0.8; for SP8 confocal microscope – NA is 0.75. We added a clarification to the manuscript.

 In light of the reviewer note we added references to enrich the discussion of our manuscript. The following paragraph was added to the discussion:

CSVD is yet an incurable disease. SBH/y-DOCA rats may serve as a novel platform for testing potential therapeutics, such as the FDA-approved drug trifluoperazine (TFP). TFP, a calmodulin inhibitor, was effectively shown to reduce cerebral edema by inhibiting aquaporin-4 (AQP4) localization [40,41]. Although the detailed molecular mechanism of brain water transport is lacking, AQP4 has been implicated as a key determinant in the glymphatic system, and therefore a potential therapeutic target for brain disorders presenting edema [42-46]. As mentioned, CSVD is characterized with various manifestations, that can be utilized as therapeutic candidates to be evaluated using evolving approaches for drug discovery and development. High throughput screening (HTS) and computer-aided drug design (CADD) are examples for such approaches, gaining interest in recent years [47,48]. The importance of these tools lies in the improved drug screening efficiency with higher predictability and clinical applicability. Nevertheless, reliable animal models, such as SBH/y-DOCA rats, are still required for testing target validation and functional efficiency.

Attached is the revised manuscript, including all changes following all of the reviewers’ comments.

Reviewer 3 Report

The study by Guy et al investigated a potential new animal model for CSVD. They found that SBH/y rats administered with DOCA and salt develop cognitive impairment and cerebrovascular pathologies associated with CSVD including ICH, BBB permeability and blood vessel stenosis. This is comprehensive and interesting study. I have a few comments that should be addressed.

  1. It looks like SBP went up by at least 40 mmHg in SBN/y DOCA group. It is not correct to interpret that SBP was not elevated in this group
  2. N's are missing for figure 1C legend
  3. Was plasma GSSG measured too? It would be good to have the GSH/GSSG ratio
  4. Figure 1F: Is mean neurological score the mean from day 32 to 57? In figure 1E the highest dsDNA value is ~500, however, in figure 1F there are values higher than this. Why aren't these values in figure 1E too? Why isn't there a neurological score for every animal?
  5. N=4 is low for immunofluorescence
  6. Figure 3J: Data does not look normally distributed
  7. Figure 4B: Date does not look normally distributed for sham
  8. Figure 5A: Can a close up image of luxol blue and cresyl violet staining for SBH/y sham be provided too for comparison?
  9. It is not clear at which timepoint the MRI analysis was performed
  10. Why are there less n's in figure 6D and E compared to figure 6C? Should the n's be the same?
  11. N=5-8 is relatively low for contextual fear conditioning test. Was this test not performed in all animals? Was a power test conducted?

Author Response

The authors would like to thank the reviewers for reading and evaluating our paper. We have addressed the major comments and revised the manuscript to the best of our abilities. Below can be found detailed reference to each comment from the reviewers.

1. It looks like SBP went up by at least 40 mmHg in SBN/y DOCA group. It is not correct to interpret that SBP was not elevated in this group

Thank you for the comment.

Although a moderate increase in SBP was observed in the SBN/y-DOCA group during the 40 days of measurement, a similar increase was also observed in the SBN/y sham group. No significant difference was observed between SBP of SBN/y-DOCA and SBN/y sham group at any time point, and therefore the moderate increase in blood pressure could not be attributed to DOCA administration.

2. N's are missing for figure 1C legend

We agree with this remark, and corrected the text accordingly. N’s were added inside the text.

3. Was plasma GSSG measured too? It would be good to have the GSH/GSSG ratio

GSH/GSSG ratio is indeed known to be reduced in neurodegenerative diseases, and can serve as an indicator for cellular health. However, there are potential pitfalls in GSSG measurements that can lead to artificial oxidation of GSH to GSSG thus generating invaluable biased GSH/GSSG ratios [1]. Fortunately, GSH depletion was already suggested by itself as not only a consequence of oxidative stress, but also as an active player in the pathogenesis of many diseases, among of which are neurodegenerative diseases [2,3]. Due to conflicts about the validity of GSSG measurement, we chose to measure GSH, which is more reliable and accurate test.

  1. Matuz-Mares D, Riveros-Rosas H, Vilchis-Landeros MM, Vázquez-Meza Glutathione Participation in the Prevention of Cardiovascular Diseases. Antioxidants 2021, 10(8), 1220. doi: 10.3390/antiox10081220.
  2. Ballatori N, Krance SM, Notenboom S, Shi S, Tieu K, Hammond Glutathione dysregulation and the etiology and progression of human diseases. Biol Chem 2009, 308(3), 191-214. doi: 10.1515/BC.2009.033.
  3. Martin HL, Teismann P. Glutathione- a review on its role and significance in Parkinson's disease. FASEB J 2009, 23(10), 3263-72. doi: 10.1096/fj.08-125443.

4. Figure 1F: Is mean neurological score the mean from day 32 to 57? In figure 1E the highest dsDNA value is ~500, however, in figure 1F there are values higher than this. Why aren't these values in figure 1E too? Why isn't there a neurological score for every animal?

Thank you for this comment.

The mean neurological score is indeed the mean score from day 32 to day 57 for each rat. A clarification was inserted in the figure’s legend.

Given the high number of values in figure 1E (n=19), the two high values in figure 1F (~1200) were calculated as significant outliers in figure 1E, and were therefore excluded from this graph. We do not claim that these values are incorrect, but that these rats exhibited especially severe phenotypes, which is consistent with their neurological score, as can be seen in figure 1F. In light of the reviewer note we decided not to exclude these values so that there would be a match to figure 1F. While these values increase the difference between the groups, they also increase the variability, which accurately describes the model (as well as the human disease).

The experiments were performed in a number of repetitions, and not all rats underwent a neurological assessment. Figure 1F contains all the rats that both underwent neurological assessment and have plasmatic dsDNA measurement.

5. N=4 is low for immunofluorescence

N=3/4 is highly acceptable for immunohistochemistry when the difference between groups is significant (e.g. [4-7]). 

4.     Shimazaki K, Yajima T, Ichikawa H, Sato T. Distribution and possible function of galanin about headache and immune system in the rat dura mater. Sci Rep 2022, 12, 5206. doi: 10.1038/s41598-022-09325-3.

5.     Horiguchi K, Fujiwara K, Yoshida S, Nakakura T, Arae K, Tsukada T, Hasegawa R, Takigami S, Ohsako S, Yashiro T, Kato T, Kato Y. Isolation and characterisation of CD9-positive pituitary adult stem/progenitor cells in rats. Sci Rep 2018, 8(1), 5533. doi: 10.1038/s41598-018-23923-0.

6.     Harada K, Fujita Y, Okuno T, Tanabe S, Koyama Y, Mochizuki H, Yamashita T. Inhibition of RGMa alleviates symptoms in a rat model of neuromyelitis optica. Sci Rep 2018, 8(1), 34. doi: 10.1038/s41598-017-18362-2.

7.     Fan H, Chen K, Duan L, Wang YZ, Ju G. Beneficial effects of early hemostasis on spinal cord injury in the rat. Spinal Cord 2016, 54(11), 924-32. doi: 10.1038/sc.2016.58.

6. Figure 3J: Data does not look normally distributed

7. Figure 4B: Date does not look normally distributed for sham

Data sets of both figure 3J and 4B passed normality test according to Kolmogorov−Smirnov test. Furthermore, all data sets were tested for normal distribution. This was added to the methods section.

8. Figure 5A: Can a close up image of luxol blue and cresyl violet staining for SBH/y sham be provided too for comparison?

Supplementary figure 3 was added, containing both a close-up image for SBH/y sham (Fig. S3A), and another representative luxol blue and cresyl violet staining of SBH/y sham and SBH/y-DOCA brains from a fronter area of the brain to further validate reproducibility (Fig. S3B).

9. It is not clear at which timepoint the MRI analysis was performed

Two months following model induction rats were sacrificed. As mentioned both in the abstract and in the methods, MRI scans were performed ex vivo, following sacrifice. We added a clarification to the manuscript.

10. Why are there less n's in figure 6D and E compared to figure 6C? Should the n's be the same?

Thank you very much for this comment. Since the behavioral testing was performed in two cycles, the wrong graph was accidently placed in the final manuscript. This was corrected inside the manuscript and we appreciate the reviewer’s keen eye to details that enabled this rectification.

11. N=5-8 is relatively low for contextual fear conditioning test. Was this test not performed in all animals? Was a power test conducted?

Every effort was made to reduce the number of rats used and minimize their suffering, and therefore the contextual fear conditioning, which is considered a relatively distressing test was not performed in all animals. Several behavioral studies indicate that five animals are adequate in order to obtain significant results [8,9].

  1. Pietersen CY, Bosker FJ, Postema F, den Boer JA. Fear conditioning and shock intensity: the choice between minimizing the stress induced and reducing the number of animals used. Lab Anim 2006, 40(2), 180-5. doi: 10.1258/002367706776319006.
  2. Wilensky AE, Schafe GE, LeDoux JE. The Amygdala Modulates Memory Consolidation of Fear-Motivated Inhibitory Avoidance Learning but Not Classical Fear Conditioning. J Neurosci 2000, 20(18), 7059-66. doi: 10.1523/JNEUROSCI.20-18-07059.2000.

Attached is the revised manuscript, including all changes following all of the reviewers’ comments.

Round 2

Reviewer 2 Report

Dear Editor,

The authors have successfully addressed the majority of my comments and concerns in order to improve the quality of the manuscript.

I believe that the new sections, improved ones, and updated references, have contributed to enhancing the clarity of the manuscript, which I can now endorse for publication.

All the best!